# Super-Resolution Microscopy Reveals Diversity of Plant Centromere Architecture

**DOI:** 10.3390/ijms21103488

**Published:** 2020-05-15

**Authors:** Veit Schubert, Pavel Neumann, André Marques, Stefan Heckmann, Jiri Macas, Andrea Pedrosa-Harand, Ingo Schubert, Tae-Soo Jang, Andreas Houben

**Affiliations:** 1Leibniz Institute of Plant Genetics and Crop Plant Research (IPK) Gatersleben, 06466 Seeland, Germany; heckmann@ipk-gatersleben.de (S.H.); schubert@ipk-gatersleben.de (I.S.); houben@ipk-gatersleben.de (A.H.); 2Biology Centre, Czech Academy of Sciences, 37005 České Budějovice, Czech Republic; neumann@umbr.cas.cz (P.N.); macas@umbr.cas.cz (J.M.); jangts@cnu.ac.kr (T.-S.J.); 3Department of Chromosome Biology, Max Planck Institute for Plant Breeding Research, 50829 Cologne, Germany; amarques@mpipz.mpg.de; 4Department of Botany, Federal University of Pernambuco (UFPE), Recife 50670-901, Pernambuco, Brazil; andrea.harand@ufpe.br; 5Department of Biological Sciences, Chungnam National University, Daejeon 34134, Korea

**Keywords:** CENH3, CENP-A, clustered centromere, *Cuscuta*, holocentromere, *Lathyrus*, *Luzula*, microtubule, monocentromere, *Pisum*, *Rhynchospora*, structured illumination microscopy

## Abstract

Centromeres are essential for proper chromosome segregation to the daughter cells during mitosis and meiosis. Chromosomes of most eukaryotes studied so far have regional centromeres that form primary constrictions on metaphase chromosomes. These monocentric chromosomes vary from point centromeres to so-called “meta-polycentromeres”, with multiple centromere domains in an extended primary constriction, as identified in *Pisum* and *Lathyrus* species. However, in various animal and plant lineages centromeres are distributed along almost the entire chromosome length. Therefore, they are called holocentromeres. In holocentric plants, centromere-specific proteins, at which spindle fibers usually attach, are arranged contiguously (line-like), in clusters along the chromosomes or in bands. Here, we summarize findings of ultrastructural investigations using immunolabeling with centromere-specific antibodies and super-resolution microscopy to demonstrate the structural diversity of plant centromeres. A classification of the different centromere types has been suggested based on the distribution of spindle attachment sites. Based on these findings we discuss the possible evolution and advantages of holocentricity, and potential strategies to segregate holocentric chromosomes correctly.

## 1. Introduction

During mitotic and meiotic cell proliferation, correct segregation of the genetic material to the daughter cells is essential. Spindle fibers attach to specific regions, called centromeres, at the highly condensed metaphase chromosomes (reviewed in [1]). Different types of centromeres have been described in yeasts, animals, and plants and are classified into two main categories: monocentromeres and holocentromeres.

The centromere size based on centromere-specific chromatin varies highly among eukaryotes, even in monocentric chromosomes. It ranges from 125 bp in the budding yeast point centromere to several Mbp in regional human centromeres [2,3,4]. At relatively large chromosomes (>2 µm), regional centromeres are localized within a distinct primary constriction cytologically visible at metaphase. Smaller chromosomes (<2 µm) often do not show such constrictions and the localization of their centromeres is difficult, as in duckweed [5] or *Genlisea* [6,7] species. Nevertheless, spindle fibers attach at a distinct chromosome region. Common bean (*Phaseolus vulgaris*) chromosomes are ~2 µm, but the primary constrictions are only visible in some chromosomes and cells [8]. The same is true for *Citrus*, but here the constrictions are visible depending on chromosome condensation and preparation [9]. In contrast to point and regional centromeres, holocentromeres are distributed almost along the entire metaphase chromosome length, and therefore do not exhibit a primary constriction [10]. Interestingly, regional centromeres with elongated constrictions may also occur in legumes [11,12,13], fire ant species [14], muntjacs [15], as well as in marsupial hybrids [16] and cancer cell lines [17].

Proteinaceous kinetochore complexes assemble at centromeres where microtubules attach for chromosome segregation. In most eukaryotes (except in budding yeast, [18]), these assembly sites are not determined by specific DNA sequences. Instead, the centromere-specific histone H3 (CENH3)/centromere protein A (CENP-A) [19] specifies the kinetochore positions in many eukaryotes and recruits additional kinetochore proteins [1]. However, no CENH3/CENP-A has been found in holocentric *Lepidoptera* and *Hemiptera* insects [20], kinetoplastids (e.g., trypanosomes, [21]) and some fungi, such as *Mucor* [22,23], *Phycomyces blakesleeanus* and *Mortierella verticillata* [24].

To decipher the structure and organization of centromeres, super-resolution microscopy, beside other microscopy techniques, has been applied. Super-resolution microscopy techniques, such as spatial structured illumination microscopy (3D-SIM), are subdiffraction imaging methods bridging the resolution gap between light and electron microscopy [25,26,27,28,29,30]. 3D-SIM allows, compared to other super-resolution microscopy techniques, fast high throughput multicolor imaging by doubling the resolution of wide-field microscopy and achieving best contrast in thin specimens [31]. Super-resolution microscopy was applied successfully in cell biology [32,33,34] to specimens from both prokaryotes and eukaryotes and allowed discovery of new structures within mammalian [35] and plant chromatin [36].

Here, we summarize findings achieved via investigating the plant ultrastructural centromere variability by 3D-SIM imaging of chromatin after immunostaining with centromere-specific antibodies.

## 2. Centromere Diversity in Plants

In recent studies with the application of 3D-SIM super-resolution microscopy and immunodetection of CENH3/CENP-A, several other centromere and (peri)centromere-specific proteins and tubulin (Figure 1, Figure 2 and Figure 3) provided the basis for creating detailed models of centromere organization in plants (Figure 4 and Table 1). Most studied plant species with large chromosomes (>2 µm) typically show a distinct primary constriction. They represent a regional monocentromere containing one cluster of CENH3/CENP-A surrounded by (peri)centromeric chromatin marked by cell-cycle dependent post-translational histone modifications, such as H2A phosphorylated at threonine 120 (H2AT120ph) [37] and histone H3 phosphorylated at serines 10 [38] and 28 [39]. However, other less common centromere-specific structures are present in plants.

A specialized monocentric organization has been found for mitotic chromosomes in *Pisum* and *Lathyrus* species [11,12,13]. In these species, primary constrictions span up to a third of the chromosome length, corresponding to 263 Mbp, and contain up to five explicit CENH3/CENP-A-containing domains. All these domains contain both CENH3/CENP-A variants (CENH3-1 and CENH3-2) identified in these species and assemble on arrays of satellite DNA with a length varying from hundreds of kilobases to almost 3 Mbp. Thus, each of these domains can be viewed as one regional centromere. Individual CENH3/CENP-A-containing domains are particularly well discernible from prophase to prometaphase, but when condensation proceeds towards metaphase, they attach to each other or even merge into line-like structures similar to holocentromeres. H2AT120ph chromatin surrounds the CENH3/CENP-A regions, but does not intermingle with them. Microtubules of the mitotic spindle attach to chromosomes at each CENH3/CENP-A-containing domain or along the entire CENH3/CENP-A-containing line-like region, indicating that CENH3/CENP-A is a faithful marker of functional kinetochores in these species (Figure 4). Although these centromeres are functionally similar to monocentromeres, they were designated as “meta-polycentromeres” to reflect the unique organization of CENH3/CENP-A-containing chromatin domains.

Line-like contiguous holocentromeres within a groove were first identified in *Luzula* (Juncaceae) species during mitosis and meiosis [41,48,49,50,53]. Later on, a similar line-like holocentromere structure and arrangement was found during mitosis in related species of the genus *Rhynchospora* (Cyperaceae) [51,54]. In both cases, line-like holocentromeres show similar localization of CENH3/CENP-A and H2AT120ph and complete association to spindle microtubules during mitotic cell divisions [41,55]. Interestingly, holocentromeres in *Rhynchospora* were also associated with centromere-specific DNA sequences (tandem repeats and centromeric retrotransposons of maize) [55], while in *Luzula*, no centromere-specific repeats have been found.

Recently, an atypical holocentromere type was identified in mitotic chromosomes of *Cuscuta europaea* [46]. In this species, the mitotic spindle microtubules attach at uniform density along the entire chromosome length, clearly indicating the holocentric nature of the chromosomes. However, CENH3/CENP-A is restricted to only one to three discrete heterochromatin bands per chromosome, and H2AT120ph is not detectable (P. Neumann, unpublished). Additionally, CENH3/CENP-A is not confined to the chromosome periphery as would be expected for a kinetochore-specific protein. Instead, it homogenously fills the entire heterochromatin band. These data suggest that CENH3/CENP-A either lost its function or acts in parallel to an additional CENH3/CENP-A-independent mechanism of kinetochore positioning [46]. Interestingly, in fire ants, the centromere expansion occurred via a mechanism independent of CENH3/CENP-A duplication [14].

## 3. Variation at Different Stages of the Cell Cycle

Centromere organization on meiotic chromosomes differs between *Rhynchospora* and *Luzula.* In *Luzula*, meiotic chromosomes maintain the CENH3/CENP-A-containing line-like holocentromeres within a groove [49]. However, in contrast to *Luzula*, *R. pubera* shows a different meiotic chromosome structure. No linear holocentromeres, but a dispersed distribution of H2AT120ph, are present [47,55] (Figure 3). CENH3/CENP-A- and CENP-C-positive centromere domains are localized exclusively at the bivalent surface. The surface of the bivalents has a similar rough structure after surface rendering as somatic prophase chromosomes and interphase nuclei. Several CENH3/CENP-A domains are present along the entire meiotic chromosomes, but as distinct clusters that specifically interact with spindle fibers. Thus, during meiosis in *Rhynchospora*, the chromosomes are holocentric according to the definition that a chromosome is holocentric when kinetochore proteins and the spindle fibers are distributed along almost the entire poleward surface of the chromatids [10,56,57]. To distinguish the centromere arrangement between mitotic and meiotic chromosomes of *R. pubera*, we name these subtypes line-like and cluster-like holocentromeres, respectively. Interestingly, the line-like holocentromere becomes reestablished during first pollen mitosis [47,58]. Similar centromere structures in both mitotic and meiotic chromosomes have been observed in other *Rhynchospora* species (A. Marques, unpublished).

During interphase in *L. elegans* and *R. pubera*, the line-like holocentromeres disperse and a high number of CENH3/CENP-A-containing centromeric units are formed [48,51,53]. The line-like metaphase holocentromeres may be the result of merging various centromeric subunits caused by chromosome condensation (Figure 3). In *L. elegans*, the centromere-specific colocalization of α-kleisin cohesin subunits with CENH3/CENP-A along the holocentromeres was proven during mitosis and meiosis [59]. This suggests that centromere-specific cohesin could act to connect the centromere units along the centromere groove.

## 4. Centromere Evolution

Holocentric chromosomes are present in green algae, protozoans, invertebrates, and various plants [10,60]. They developed repeatedly via convergent evolution [20,56,60,61,62,63,64]. All known holocentric plants are monocot (species of Melanthiaceae, Juncaceae, and Cyperaceae) or eudicot (species of *Drosera* and *Cuscuta* subg. *Cuscuta*) angiosperms [20,60,65]. Even within the same genus, such as *Cuscuta*, mono- (*C. japonica*) and holocentric (*C. europaea*) species may appear [46]. Such closely related species offer the possibility for comparative studies to analyze the molecular basis of centromere type variability. However, the factors that induced the transition from mono- to holocentromeres and the underlying mechanisms are still under debate. The surprisingly high centromere diversity in independent eukaryotic lineages raises the questions of whether this variability offers evolutionary advantages and how holocentromeres evolved from monocentric ones or vice versa. Additionally, it is of interest to investigate how the different centromere types guarantee proper chromosome segregation during mitosis and meiosis.

### 4.1. Is Holocentricity Original or Derived?

Lima de Faria [66] postulated that mono- and holocentromeres may have derived from distinct groups of special chromomeres that are either restricted to a particular site or uniformly dispersed on mitotic chromosomes. Other authors proposed that holocentricity is a primitive feature, because it has been found in phylogenetically basal plant and animal taxa and offers fast evolution combined with the high tolerance of chromosomal fragmentation and rearrangements [67,68,69,70,71,72,73]. Sybenga [71] suggested that monocentricity developed from holocentricity. In contrast, due to the presence of holocentricity in phylogenetically distant and derived taxa, such as holocentric hemipterans and rushes (Juncaceae), which descended from more primitive monocentric species, Swanson [74] hypothesized that holocentricity has been derived from monocentricity. Greilhuber [75] and Melters et al. [60] supported this hypothesis by arguing that holocentric chromosomes developed four times independently in angiosperms. Nagaki et al. [53] suggested that in *Luzula* holocentricity may have arisen from monocentricity via subsequent centromere extension, and Neumann et al. [11] proposed that centromere-competent satellite(s) caused the transition from mono- to holocentric chromosomes based on the findings in extended *Pisum* centromeres. The “telomere to centromere” model of Villasante et al. [76,77] and the “centromere drive” hypothesis of Malik and Henikoff [78] agree with the hypothesis that holocentricity evolved from monocentric chromosomes (Figure 5).

Similar to *Pisum* and *Lathyrus* species [11,12,13], the red fire ant, *Solenopsis invicta*, and closely related species possess elongated centromeres. Consequently, they are regarded as evolutionary intermediates via runaway expansion of their centromeres towards the development of holocentromeres [14]. Our recent ultrastructural centromere investigations, which highlight the variation of centromere organization in holocentric plants, support the view that holocentricity may have been derived from monocentricity independently multiple times.

The occurrence of increased exposure to clastogens (factors inducing chromosome fragmentation), such as cosmic irradiation during terrestrialization (land colonization), may have caused the convergent evolution of holocentricity in different eukaryotic taxa [79]. Král et al. [80] found that holocentricity in spiders is an autapomorphy (a derived trait unique to a given taxon) of the superfamily *Dysderoidea*. It may have developed via multiple subsequent chromosome fusions of monocentric chromosomes combined with genome size reduction. But, considering the unstable nature of dicentric chromosomes [81], simple fusion of monocentric chromosomes is less likely to be the route towards holocentric chromosomes.

### 4.2. Is Holocentricity Related to CENH3/CENP-A Loss?

Spindle fibers commonly attach to kinetochores organized by CENH3/CENP-A-containing regions of centromeres. However, recent studies showed that a number of insects have recurrently lost CENH3/CENP-A, implying that they evolved a CENH3/CENP-A-independent mechanism of kinetochore assembly. Interestingly, all of them have holocentric chromosomes, derived independently from monocentric ones. Despite the loss of CENH3/CENP-A in holocentric insects, these holocentromeres still comprise kinetochore proteins that can force apart the chromosomes [20]. Thus, the transition to holocentricity may facilitate the loss of CENH3/CENP-A, at least in some species, or vice versa [20,82]. Since the mitotic spindle microtubuli attach to chromosomes at sites devoid of CENH3/CENP-A in *C. europaea*, it is likely that a CENH3/CENP-A-independent pathway of kinetochore assembly evolved also in this species [46].

A hybrid-type centromere has been discovered in the pathogenic fungus *Mucor circinelloides*. This CENH3/CENP-A-lacking centromere type displays a mosaic of point and regional centromeres [22,23].

It can be speculated that the loss of CENH3/CENP-A and the variation of the CENH3/CENP-A content might cause the observed centromere plasticity. Alternatively, structural changes in the centromere architecture might have enabled CENH3/CENP-A-independent centromere activity [57].

## 5. Advantages and Challenges of Holocentricity

One reason that holocentricity evolved independently in different taxa of eukaryotes could be that chromosome fragmentation by DNA breakage seems to be more easily tolerated in species with holocentric chromosomes [79,83]. Contrary to monocentric species where acentric fragments usually become lost during cell division, the breakage of holocentric chromosomes creates fragments with normal spindle fiber attachment sites [84,85]. The fast formation of new telomeres at the break points allows holocentric species a rapid karyotype evolution, including chromosome fissions and rearrangements [57,86] (Figure 6). A high karyotype variability between related holocentric species, e.g., within the genera *Carex* and *Eleocharis*, with chromosome numbers of 2*n* = 12–124 and 2*n* = 6–196, respectively, probably reflects this situation [87]. In addition, a chromosome number reduction to *x* = 3 from the possible basic number of *x* = 5 occurred in *E. subarticulata* [88]. An intraspecific chromosome number variability with 2*n* = 41–47 in *E. kamtschatica* [89] and 2*n* = 6–8 in *E. maculosa* [90] has been reported. On the other hand, a comparison of all known holocentric lineages with their closest related monocentric lineages revealed that the different speciation rates between mono- and holocentrics are not related to different centromere types [91].

Could it be that the development of holocentricity is advantageous to move large chromosomes during cell divisions? Obviously, this is not a reason because, similar to the large chromosomes of *Luzula* [41] and *Rhynchospora* [47,51], the much smaller chromosomes of *Cuscuta* [46], *Drosera* [92,93,94,95] and *Chionographis* [96,97] are holocentric, and the holokinetic chromosomes of the spider superfamily *Dysderoidea* are of different size [80]. On the other hand, the very large chromosomes of lilies, of *Triticeae*, of some legumes, and of conifers, for instance, are monocentric. Generally, holocentric species have smaller genomes compared to their monocentric relatives [87].

In contrast to possible evolutionary advantages, holocentric chromosomes might represent a challenge during nuclear division, because due to the extended centromere architecture, a merotelic attachment of spindle fibers, which may lead to chromosome mis-segregation, is more likely [57,60]. As a consequence, holocentric organisms had to adapt their meiotic processes [98].

Holocentric animals adapted the meiotic chromosome segregation by remodeling their chromosomes into functionally monocentric ones, as in the worm *Caenorhabditis elegans*, by adopting a telokinetic behavior as in *Heteroptera* bugs, and by “inverted meiosis” as in the citrus mealybug *Planococcus citri* [49]. The holokinetic plant species *L. elegans* [49,50], *R. pubera* and *R. tenuis* [55,99] display an inverted order of meiotic chromatid segregation. Sister chromatids separate already during meiosis I, while the homologues segregate in meiosis II [50].

## 6. Conclusions

Here, we show that centromere-specific antibodies combined with super-resolution microscopy are useful to identify the ultrastructure of the different centromere types and microtubule attachment sites. Recent analysis of species with nonclassical centromeres has demonstrated that in addition to monocentromeres, meta-polycentric chromosomes, cluster-, and line-like holocentromeres developed during plant evolution. The ultrastructural centromere investigations showing interspecific centromere variability, and in some species variability between mitotic and meiotic centromeres, suggest a development from monocentromeres towards holocentromeres. An opposite direction of evolution or independently from an original monocentromere organization is less likely. Whether meta-polycentromeres are an intermediate between mono- and holocentromeres is not confirmed, because until now, no lineage that contains species with both types of centromeres, meta-polycentric and holocentric, has been described. The centromere unit-based organization of holocentromeres is dynamic, thus allowing the formation of line- or cluster-like holocentromeres.

Based on the distribution of spindle attachment sites at CENH3/CENP-A-containing or CENH3/CENP-A-free regions, a classification of different centromere types and subtypes has been suggested. No correlation exists between holocentricity and chromosome size. Holocentricity might be advantageous due to a higher tolerance of chromosome fragmentations.

Alternative mechanisms have developed to segregate holocentric chromosomes, showing that proper chromosome segregation during mitosis and meiosis may be performed not only by distinct monocentromeres. Elongated centromeres and the different types of holocentromeres also fulfill this essential task successfully.

## Figures and Tables

**Figure 1 ijms-21-03488-f001:**
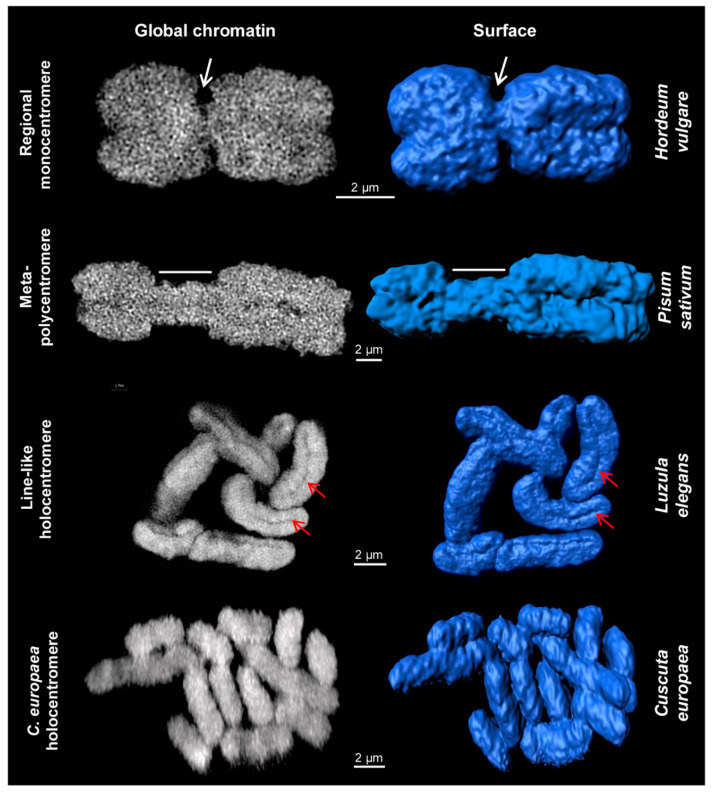
Different centromere types of somatic plant metaphase chromosomes. Images were obtained via global chromatin labelling by DAPI. After surface rendering of structured illumination microscopy (SIM) image stacks [40] using the Imaris 8.0 software, the centromere structure variability of different plant species becomes visible. Regional monocentromeres are characterized by a district primary constriction (white arrows). Meta-polycentromeres represent an elongated primary constriction (region indicated by dashes). Line-like holocentromeres are characterized by the arrangement of centromere-specific proteins in a distinct line within a groove (red arrows), as found in *Luzula* and *Rhynchospora* (Figure 3). Holocentromeres in *Cuscuta europaea* are structures where spindle fibres attach along the whole chromosome at centromere-specific histone H3 (CENH3)/centromere protein A (CENP-A)-chromatin as well as at CENH3/CENP-A-free regions (see also Figure 2 and Figure 4), but the surface is relatively smooth without a specific constriction.

**Figure 2 ijms-21-03488-f002:**
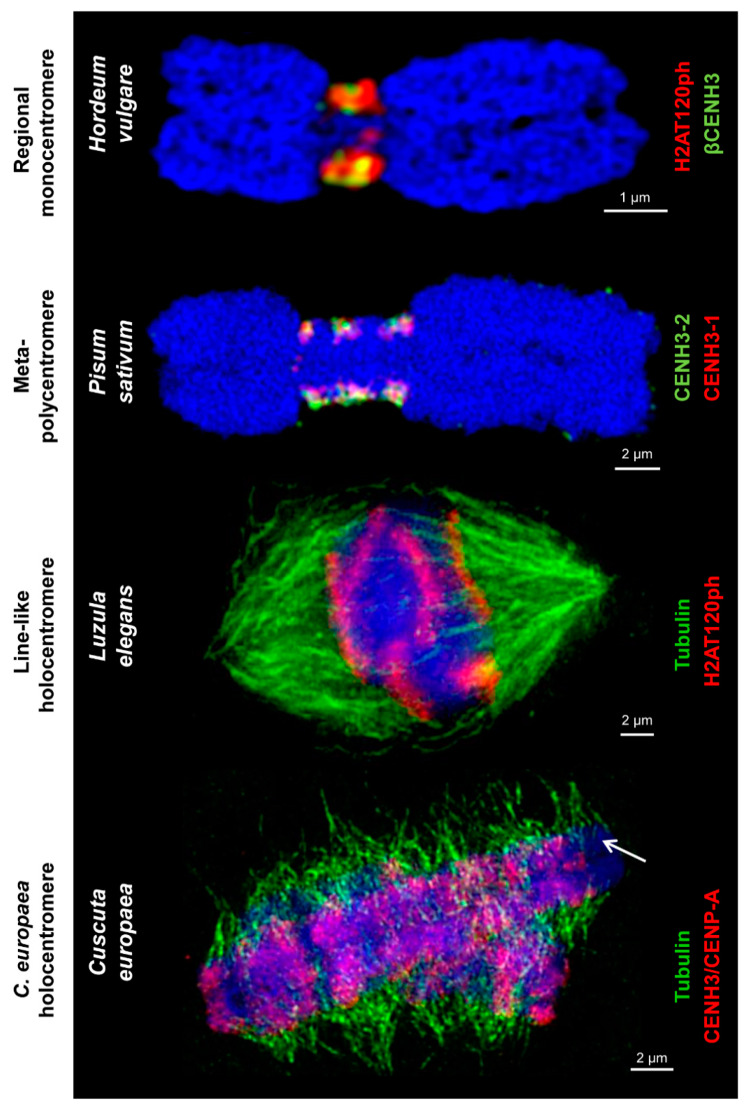
Different centromere types labeled by centromere-specific histone markers and tubulin. These markers, such as different CENH3/CENP-A variants and H2A phosphorylated at threonine 120 H2AT120ph, intermingle in regional monocentromeres. Spindle fibers attach to H2AT120ph-containing regions of line-like *Luzula* holocentromeres and CENH3/CENP-A-containing and CENH3/CENP-A-free regions of *C. europaea* holocentromeres, respectively, along the entire chromosomes. The arrow marks chromosome 1 of *C. europaea* with a chromosome-wide distribution of tubulin and restricted amount of CENH3/CENP-A. Chromosomes are counterstained with DAPI (in blue).

**Figure 3 ijms-21-03488-f003:**
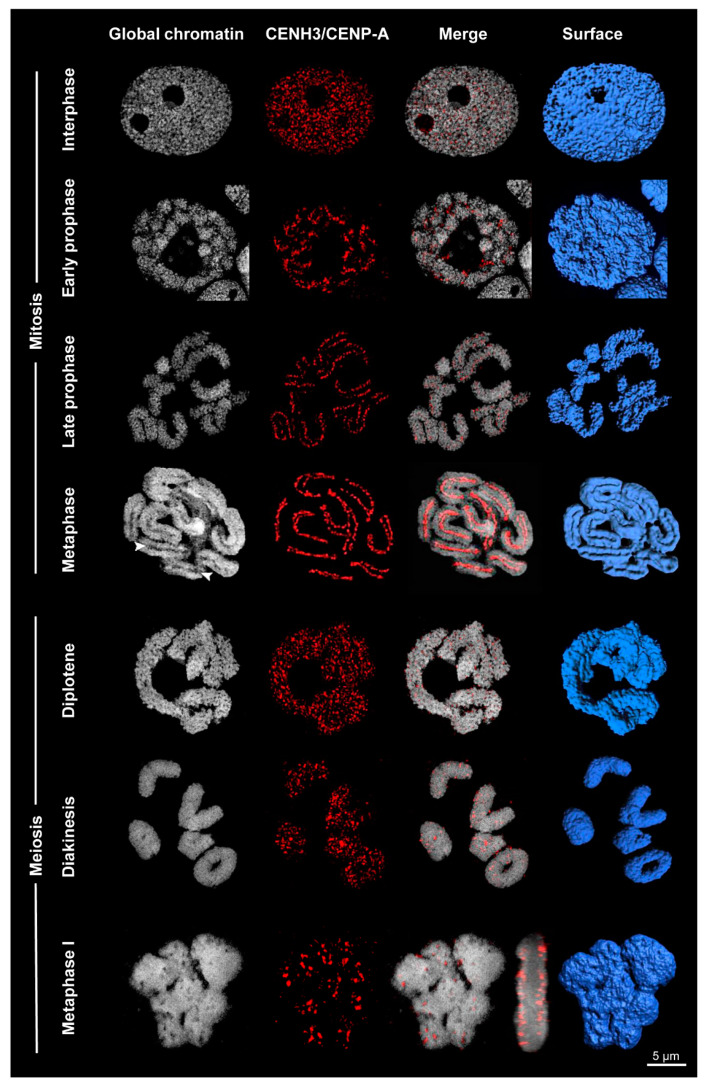
Centromere formation differs between mitosis and meiosis of *Rhynchospora pubera.* Whereas line-like holocentromeres appear in mitosis, cluster-like holocentromeres become established in meiosis. The process of global chromatin condensation and the dynamics of CENH3/CENP-A arrangement is visualized by DAPI staining and immunolabeling with CENH3/CENP-A-specific antibodies. Surface rendering of SIM image stacks clearly indicates the presence of grooves (arrowheads) at somatic metaphase chromosomes, but their absence at metaphase I bivalents. The merged side-view of the metaphase I cell reveals CENH3/CENP-A at the surface, but not inside the bivalents.

**Figure 4 ijms-21-03488-f004:**
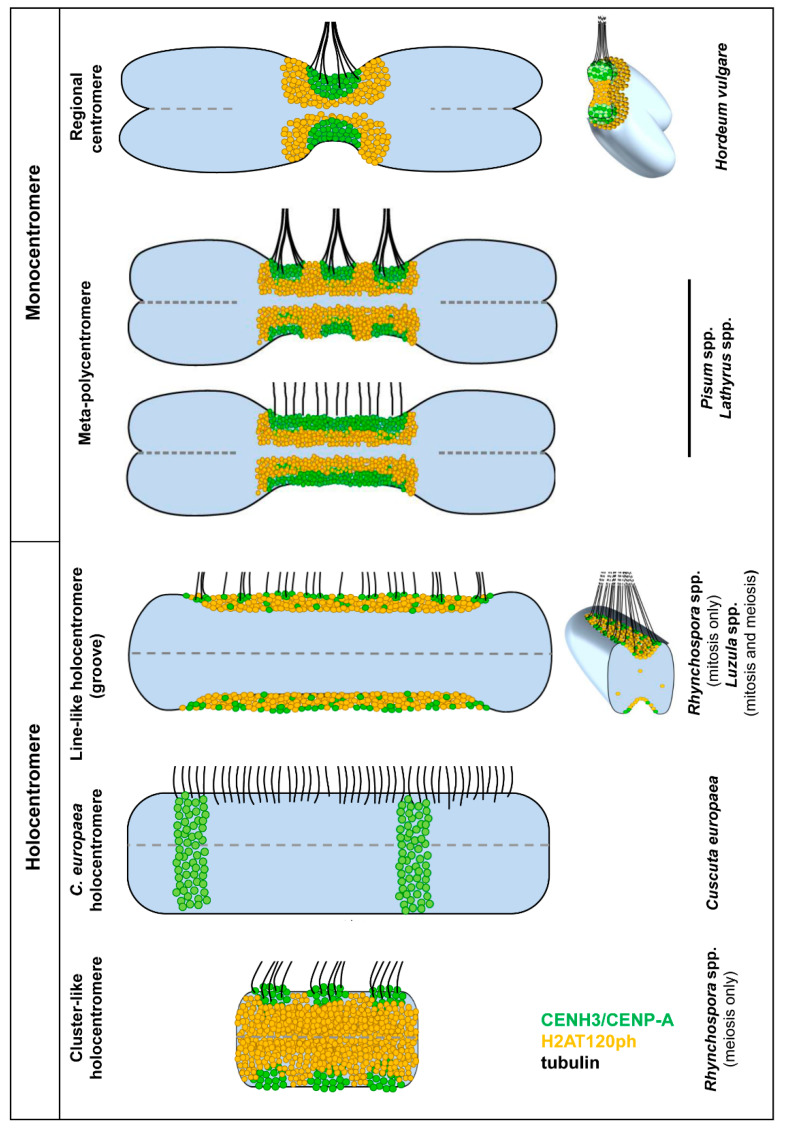
Models of the different mono- and holocentromere types appearing in different plant species indicate the possible centromere plasticity during mitosis and meiosis. The classification is based on the distribution of the spindle fibre attachment sites. In mono- and meta-polycentromeres, the microtubules (tubulin) form branching bundles and attach mainly at the flanks of the CENH3/CENP-A clusters, but not at H2AT120ph. The bundle formation is less pronounced at holocentromeres. In line-like holocentromeres, spindle fibres attach mainly as single microtubules at the rim along the entire groove containing CENH3/CENP-A and H2AT120ph, as is clearly visible in the cross-section [41]. The CENH3/CENP-A-containing domains in meta-polycentromeres are usually well discernible (the upper model), but may also fuse into one line-like domain (bottom model). In *C. europaea* holocentromeres, the spindle fibres also attach to CENH3/CENP-A-free chromatin.

**Figure 5 ijms-21-03488-f005:**
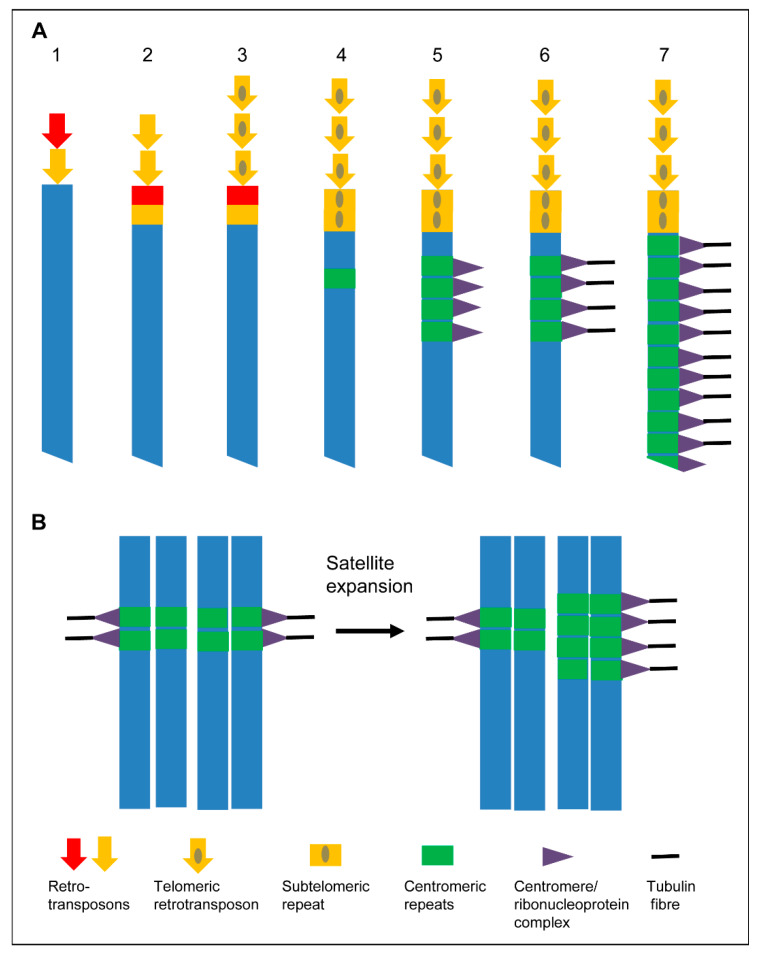
Models to expand centromeres. (**A**) “Telomere to centromere” model based on Villasante et al. [76,77]. (1) Different retroelements are mobilized to heal the DNA ends of a broken chromosome. (2) The most effective telomeric retrotransposons become selected. (3) Retrotransposons with a more effective capping capability are selected. (4) Subtelomeric repeats appear. (5) Subtelomeric repeats are amplified and become centromeric repeats. Ribonucleoprotein complexes are formed after transcription of theses repeats. (6) Centromeric repeats become a protocentromere after being recognized by microtubules. (7) Chromosome wide extension of the centromere to form a holocentromere. (**B**) “Centromere drive model” of Henikoff et al. [2]. The expansion of a satellite that binds CENH3/CENP-A provides more microtubule attachment sites. This stronger centromere drives in female meiosis into the egg cell.

**Figure 6 ijms-21-03488-f006:**
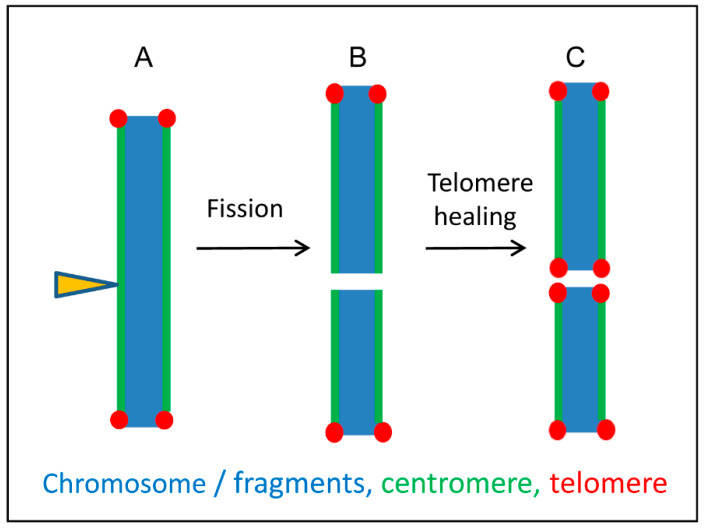
A model illustrating possible karyotype alterations after fragmentation of holocentric chromosomes based on the interplay between holocentricity and telomere healing. (**A**) Irradiation of holocentric chromosomes (arrowhead) induces chromosome fragments. Centromere activity is present along all chromosome fragments. (**B**) Broken ends lack telomere repeats and become gradually healed (**C**) by de novo telomere syntheses. Based on Jankowska et al. [86].

**Table 1 ijms-21-03488-t001:** Centromere types present in yeast and plant species classified based on the distribution of spindle fiber attachment sites.

Centromere Type	Centromere Subtype	Features	Species
**Monocentromere**	**Point centromere ***	Single CENH3/CENP-A-containing nucleosome.	Budding yeast [42]
	**Regional monocentromere ***	Single CENH3/CENP-A-containing chromatin domain where mitotic spindle fibers attach. This domain is mostly located in the primary constriction if it is discernible.	Fission yeast [43], e.g., *Hordeum vulgare* [37,41,44], *Secale cereale* [45], *Cuscuta japonica* [46]
	**Meta-polycentromere**	Elongated primary constriction possessing 2-5 CENH3/CENP-A-containing chromatin domains where spindle fibers attach.	*Pisum sativum* [11,12,13], *Lathyrus* [12,13]
**Holocentromere**	**Cluster-like holocentromere**	Many evenly dispersed CENH3/CENP-A-clusters where spindle fibers attach along the whole chromosome without a groove.	*Rhynchospora pubera* (meiosis) [47]
	**Line-like holocentromere**	Many CENH3/CENP-A-containing chromatin domains forming a contiguous line along the whole chromosome. Spindle fibers attach at CENH3/CENP-A-positive chromatin along a groove.	*Luzula elegans* [37,41,48,49,50], *L. luzuloides* [37], *R. pubera* (mitosis) [47,51], *R. tenuis* (mitosis)
	**Holocentromere in *C. europaea***	Attachment of mitotic spindle fibers along the entire chromosome length, which does not correlate with the distribution of CENH3/CENP-A. It is not yet clear which proteins constitute the centromere in this species.	*C. europaea* [46]

* This classification is based on the amount of CENH3/CENP-A-positive chromatin. However, from a cell biology perspective, in yeasts there is a high similarity between point and regional centromeres regarding their structures and kinetochore separation distances [52].

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
