# Peer review of "Super-Resolution Microscopy Reveals Diversity of Plant Centromere Architecture"

_ijms, 2020, doi:10.3390/ijms21103488_

Round 1
Reviewer 1 Report
The manuscript entitled “Super-resolution microscopy reveals diversity of
plant centromere architecture” by Shubert et al. is a review about the structural variation of the centromere in plants. They used information from centromer-specific antibodies and super-resolution microscopy reveiling different centromere types and sites for microtubules attachment.
It is very well written and presents a good functional and evolutionary perspective of plant centromers in a fine-scale. I found the manuscript very useful for many researchers working in the topic and for this I recommend it for publication in IJMS.
Author Response
Dear reviewer 1,
thank you very much for your positive evaluation.
Reviewer 2 Report
“Super-resolution microscopy reveals diversity of 2 plant centromere architecture” by Schubert et al.
This manuscript is a comprehensive overview of recent findings about the ultrastructure of the different centromere types by using super-resolution microscopy with centromere-specific antibodies in plants. Novel insights about plant evolution in centromere structure such as monocentromeres, meta-polycentric, cluster- and line-like holocentromeres are described. Thus, I recommend this review for publication after one issue below is fixed.
The authors use the word “CENPH3” to describe the centromere-specific histone H3 variant CENH3 which specifies the kinetochore positions in many eukaryotes. Although specific gene/protein names are used in each organism, there is a consensus in the centromere field to use “CENP-A” instead of “CENPH3” at least for describing the general function of the gene/protein over species (see the reference below). I strongly recommend using CENP-A or CENP-A homologs to describe the general function or the homologs in well-characterized centromere organisms. When “CENPH3” needs to be used, it should be “CENPH3/CENP-A”.
“CENPH3” and “CENP-A” should be added to Keywords on the front page.
Chromosome Res. 2013 Apr;21(2):101-6. doi: 10.1007/s10577-013-9347-y. Epub 2013 Apr 12. Esperanto for histones: CENP-A, not CenH3, is the centromeric histone H3 variant.
Author Response
The authors use the word “CENPH3” to describe the centromere-specific histone H3 variant CENH3 which specifies the kinetochore positions in many eukaryotes. Although specific gene/protein names are used in each organism, there is a consensus in the centromere field to use “CENP-A” instead of “CENPH3” at least for describing the general function of the gene/protein over species (see the reference below). I strongly recommend using CENP-A or CENP-A homologs to describe the general function or the homologs in well-characterized centromere organisms. When “CENPH3” needs to be used, it should be “CENPH3/CENP-A”.
We apply “CENH3/CENP-A” as suggested.
“CENPH3” and “CENP-A” should be added to Keywords on the front page.
Done.
Chromosome Res. 2013 Apr;21(2):101-6. doi: 10.1007/s10577-013-9347-y. Epub 2013 Apr 12. Esperanto for histones: CENP-A, not CenH3, is the centromeric histone H3 variant.
We added this reference.
Reviewer 3 Report
Review of Schubert et al.,
This is nice review of centromere structure in a diverse spectrum of plants. I found the manuscript easy to read and informative.
There are a few issues that should be addressed.
Figures 1, 2 and need scale bars. This raises a more interesting question. The authors discuss the canonical point vs regional centromeres in Table 1. However I suggest the authors look at this Table from Lawrimore and Bloom (Crit Rev Biochem Mol Biol. 2019 Aug;54(4):352-370. doi: 10.1080/10409238.2019.1670130. Epub 2019 Oct 1.) in which those authors suggest that from the cell biology perspective, there is more similarities in the point vs. regional centromeres than examination of the sequence. What are the sizes of the centromeres in all of the images shown herein?
I was confused by the line like holocentromeres. They say in the Table 1 that that the line inside the groove is where spindle fibers attach (Table 1 line-like holocentromere). However in Fig. 4 they don’t give an example of the microtubules in the “groove” like holocentromeres. Furthermore, I looked at several of the references given in Table 1 and while I could find the centromere staining, I was unable to find a reference for microtubules inside the groove. This information needs to be clearly referenced in the manuscript.
The only other point is the following. Again, this is based on the information illustrated in the inset herein. One perspective is that the budding yeast and mammalian centromeres are more or less a cylinder on the surface of the chromosome. Couldn’t the holocentric centromere be a filleted regional centromere. Meaning instead of a cylinder, the cylinder is unrolled, and the centromere extends to various extents along the outer edges of the chromosomes, as depicted in Figure. 4.
Author Response
Figures 1, 2 and need scale bars. This raises a more interesting question. The authors discuss the canonical point vs regional centromeres in Table 1. However I suggest the authors look at this Table from Lawrimore and Bloom (Crit Rev Biochem Mol Biol. 2019 Aug;54(4):352-370. doi: 10.1080/10409238.2019.1670130. Epub 2019 Oct 1.) in which those authors suggest that from the cell biology perspective, there is more similarities in the point vs. regional centromeres than examination of the sequence. What are the sizes of the centromeres in all of the images shown herein?
Thank you very much for the hint on the paper of Lawrimore and Bloom (2019). We added this information below table 1:
“This classification is based on the amount of CENH3/CENP-A-positive chromatin. However, from the cell biology perspective, in yeasts there is a high similarity between point and regional centromeres regarding their structures and kinetochore separation distances (Lawrimore and Bloom 2019)”.
To show the centromere sizes we added bars in figures 1 and 2 as requested.
I was confused by the line like holocentromeres. They say in the Table 1 that that the line inside the groove is where spindle fibers attach (Table 1 line-like holocentromere). However in Fig. 4 they don’t give an example of the microtubules in the “groove” like holocentromeres. Furthermore, I looked at several of the references given in Table 1 and while I could find the centromere staining, I was unable to find a reference for microtubules inside the groove. This information needs to be clearly referenced in the manuscript.
We clarified the structural situation by adding additional information, citing the reference Wanner et al. (2015), and adding images of cross-sections for a line-like holocentromere in comparison to a regional monocentromere in fig. 4.
The only other point is the following. Again, this is based on the information illustrated in the inset herein. One perspective is that the budding yeast and mammalian centromeres are more or less a cylinder on the surface of the chromosome. Couldn’t the holocentric centromere be a filleted regional centromere. Meaning instead of a cylinder, the cylinder is unrolled, and the centromere extends to various extents along the outer edges of the chromosomes, as depicted in Figure. 4.
We do not think that a holocentromere is a filleted regional centromere via unrolling a cylinder, because a different number of similar CENH3/CENP-A clusters appear at meta-polycentromeres representing elongated regional centromeres, and may be an intermediate towards the development of holocentromeres. In addition, line- and cluster-like holocentromeres originate from multiple centromeric subunits distributed in interphase nuclei (Marques et al. 2015, 2016).
In plants (at least in barley) kinetochores appear as CENH3/CENP-A-positive chromatin forming distinct “pad-like” regions at each chromatid where spindle fibers attach during metaphase (Wanner et al. 2015) rather than forming a cylinder.
Marques et al. (2015). Holocentromeres in Rhynchospora are associated with genome-wide centromere-specific repeat arrays interspersed among euchromatin. Proc Natl Acad Sci U S A 112, 13633-13638
Marques et al. (2016b). Restructuring of holocentric centromeres during meiosis in the plant Rhynchospora pubera. Genetics 204, 555-568.
Wanneret al. (2015) The ultrastructure of mono- and holocentric plant centromeres: an immunological investigation by structured illumination microscopy and scanning electron microscopy. Chromosoma 124, 503-517.
Reviewer 4 Report
This is a nice review describing plant centromere architecture. Although there are many reviews on animal centromeres, there are not so many reviews on plant centromeres. Therefore, this would be good to learn plant centromeres. This would be acceptable, but I have several comments, which will be helpful for authors to revise it before publication.
- Authors used a term “CENH3” in entire text. Although I know that researchers in the plant centromeres often use this term. But CENH3 is CENP-A and CENP-A is commonly used. To raise visibility of this review, author should use both CENH3 and CENP-A like CENH3 (CENP-A).
- In Line56-59, authors described centromeres lacking CENH3. But they did not mention the centromere in Mucor, here. Although I noticed this reference in the later part, they should cite this reference in Introduction.
- Authors introduced centromere evolution especially on holocentricity. This is an important part, but it is a bit hard to follow the text. For example, authors described “telomere to centromere” and “centromere drive” model. It would be good to add some Figures explaining such models. I am also curious about evolutional advantage for holocentromeres. If authors added a Figure for this, it would be good.
- In some holecentromeres of insect lacking CENH3, CENP-T may function instead of CENH3 (CENP-A). Authors should mention it, and introduce about centromere proreins in plant holocentromeres.
Author Response
This is a nice review describing plant centromere architecture. Although there are many reviews on animal centromeres, there are not so many reviews on plant centromeres. Therefore, this would be good to learn plant centromeres. This would be acceptable, but I have several comments, which will be helpful for authors to revise it before publication.
- Authors used a term “CENH3” in entire text. Although I know that researchers in the plant centromeres often use this term. But CENH3 is CENP-A and CENP-A is commonly used. To raise visibility of this review, author should use both CENH3 and CENP-A like CENH3 (CENP-A).
We regarded this comment by adding the citation Earnshaw et al. (2013) and applying the term CENH3/CENP-A as reviewer 1 suggested.
- In Line56-59, authors described centromeres lacking CENH3. But they did not mention the centromere in Mucor, here. Although I noticed this reference in the later part, they should cite this reference in Introduction.
We added this reference to Introduction.
- Authors introduced centromere evolution especially on holocentricity. This is an important part, but it is a bit hard to follow the text. For example, authors described “telomere to centromere” and “centromere drive” model. It would be good to add some Figures explaining such models. I am also curious about evolutional advantage for holocentromeres. If authors added a Figure for this, it would be good.
We prepared additional figures to explain the “telomere to centromere” and “centromere drive” models (Figure 5) and the evolutionary advantages of holocentromeres (Figure 6).
- In some holecentromeres of insect lacking CENH3, CENP-T may function instead of CENH3 (CENP-A). Authors should mention it, and introduce about centromere proreins in plant holocentromeres.
We added the following sentence: “Despite the loss of CENH3/CENP-A in holocentric insects, these holocentromeres still comprise kinetochore proteins that the centromere machinery can force apart the chromosomes (Drinnenberg et al. 2014)”.
Our knowledge about the kinetochore composition in holocentric plants is still limited. Therefore, we cannot add additional information on this topic.